# Radiomics and Artificial Intelligence Can Predict Malignancy of Solitary Pulmonary Nodules in the Elderly

**DOI:** 10.3390/diagnostics13030384

**Published:** 2023-01-19

**Authors:** Stefano Elia, Eugenio Pompeo, Antonella Santone, Rebecca Rigoli, Marcello Chiocchi, Alexandro Patirelis, Francesco Mercaldo, Leonardo Mancuso, Luca Brunese

**Affiliations:** 1Thoracic Surgery Unit, Policlinico Tor Vergata, 00133 Rome, Italy; 2Department of Medicine and Health Sciences V. Tiberio, University of Molise, 86100 Campobasso, Italy; 3Department of Diagnostic Imaging and Interventional Radiology, University of Tor Vergata, 00133 Rome, Italy

**Keywords:** solitary pulmonary nodule, radiomics, artificial intelligence analysis, machine learning, lung cancer, elderly

## Abstract

Solitary pulmonary nodules (SPNs) are a diagnostic and therapeutic challenge for thoracic surgeons. Although such lesions are usually benign, the risk of malignancy remains significant, particularly in elderly patients, who represent a large segment of the affected population. Surgical treatment in this subset, which usually presents several comorbidities, requires careful evaluation, especially when pre-operative biopsy is not feasible and comorbidities may jeopardize the outcome. Radiomics and artificial intelligence (AI) are progressively being applied in predicting malignancy in suspicious nodules and assisting the decision-making process. In this study, we analyzed features of the radiomic images of 71 patients with SPN aged more than 75 years (median 79, IQR 76–81) who had undergone upfront pulmonary resection based on CT and PET-CT findings. Three different machine learning algorithms were applied—functional tree, Rep Tree and J48. Histology was malignant in 64.8% of nodules and the best predictive value was achieved by the J48 model (AUC 0.9). The use of AI analysis of radiomic features may be applied to the decision-making process in elderly frail patients with suspicious SPNs to minimize the false positive rate and reduce the incidence of unnecessary surgery.

## 1. Introduction

Lung cancer is the leading cause of cancer-related death worldwide, and accounts for 23% of all deaths from malignancy [1]. Despite improvements in the diagnosis and treatment of lung cancer, the overall cure rate is still about 10%, although the 5-year survival in early-stage disease may be as high as 92% [2]. In this context, the improved overall survival has to be mainly attributed to the enhanced recognition of clinically silent solitary pulmonary nodules (SPNs) which have eventually proved to be malignant, which has been facilitated by the increased use of low-dose computed tomography (CT) and hybrid imaging techniques such as positron emission tomography–computed tomography (PET-CT) [3,4,5].

Despite the refinement of medical treatments, lung resection is still considered the best curative treatment for early-stage lung cancer [6]. Therefore, achieving prior indications of the nature of the SPN must be a priority in regard to offering a surgical chance to patients deemed to have a malignant nodule, particularly when dealing with high-risk individuals such as the elderly with associated comorbidity and functional impairment, in whom the decision to operate requires a careful balance of the risk-to-benefit ratio.

In these instances, the practice of obtaining pre-operative biopsies can be time-consuming and questionable. Quite often, CT-guided needle biopsy is challenging due to the location and the small diameter or sub-solid features of the nodule, and the risk of procedure-related complications such as hemorrhage or pneumothorax, combined with the high rate of false negatives, should be carefully considered, particularly in the elderly population [7]. In light of the recent technological developments and the relevant digitalization process, artificial intelligence (AI) is increasingly becoming an integral part of the modern approach to undetermined lesions in clinical practice.

On the other hand, radiomics is a method that involves extracting a large number of features from medical images using data-characterization algorithms. These features, termed radiomic features, have the potential to uncover tumor patterns and characteristics that are not visible to the human eye. Through the acquisition and polyparametric processing of data, the algorithm can differentiate normal images from pathological images with a high sensitivity [8].

The role of radiomics combined with AI in preoperative evaluations of SPN is becoming more and more essential [9,10,11].

We reasoned that including radiomic data in machine learning algorithms might allow for the evaluation of the predictive value of the extracted features and consequently help to identify malignant SPNs. The aim of this study was to demonstrate the validity of radiomics and machine learning as predictive and non-invasive techniques in SPN analysis, especially in elderly patients with multiple comorbidities, who require a tailored surgical approach.

## 2. Materials and Methods

### 2.1. Patients Selection

In this study, CT and PET-CT scans from 71 consecutive patients over 75 years old undergoing resection of a SPN from 2016 to 2021 were anonymously assessed in a retrospective analysis.

All the evaluated CT scans included at least one reconstruction with a slice thickness ≤1.5 mm, in accordance with the revised Fleischner Society guidelines [12].

All images with either an incidentally detected SPN or an SPN detected through screening were included. Patients with clinically advanced disease were excluded.

The clinical data collected from medical records included sex, age, smoking history, comorbidities, pre-operative vital signs and laboratory tests, type of surgery, post-operative complications, days of hospitalization and final histology.

Comorbidities were divided into cardiac (previous coronary artery disease, previous cardiac surgery, current treatment for cardiac failure, hypertension or arrhythmia) and other (insulin dependent diabetes, serum creatinine >2 mg/dl, previous cerebral vascular accident, chronic kidney failure, chronic obstructive pulmonary disease, gastric ulcer, liver disease, previous malignancy and others). The clinical data were used to calculate the Charlson comorbidity index according to different risk scores of post-operative mortality and morbidity, such as the Thorascore and POSSUM scoring system, applied in thoracic surgery [13,14,15].

### 2.2. The Radiomic Feature Set

Radiomic features have the potential to uncover disease characteristics that fail to be appreciated by the naked eye. The idea behind radiomics is that the distinctive imaging features of different forms of disease may be useful for predicting prognoses and therapeutic responses for various conditions, thus providing valuable information for tailored therapy.

The effectiveness of 71 different radiomic features in predicting the presence of pulmonary nodules was explored. These features belonged to five different categories, described as follows.

*First-order features*: this category described the distribution of voxel intensities within the region of interest (ROI), which in this study was related to the areas in the magnetic resonance image (MRI) associated with cancer. One feature belonging to this category was considered.*Shape*: this feature category included descriptors of the three-dimensional size and shape of the ROI. These features were independent of the gray level intensity distribution in the ROI and therefore were only calculated on the basis of non-derived images and masks. Fourteen features belonging to this category were considered.*Gray-Level Co-occurrence Matrix* (GLCM): this category considered the spatial relationships of pixels in the gray-level co-occurrence matrix, i.e., the gray-level spatial dependence matrix. The GLCM functions characterized the texture of an image by computing how often pairs of pixels with specific values and in a specified spatial relationship occurred in an image, and then extracting measures from this matrix. Twenty-four different features belonging to this category were considered.*Gray-Level Run Length Matrix* (GLRLM): this category was related to the size of homogeneous runs for each gray level. It quantified gray-level runs, which were defined as the length (expressed as a number of pixels) of consecutive pixels that had the same gray-level value. Sixteen features belonging to this category were considered.*Gray-Level Size Zone Matrix* (GLSZM): the features of this category were used to quantify gray-level zones in an image. A gray--level zone was defined as the number of connected voxels that shared the same gray-level intensity. A voxel was considered connected if the distance was one according to the infinity norm. Sixteen different features were considered from this category.

The full set of 71 radiomic features considered in this study is reported in Table 1.

### 2.3. The Classification Process

After obtaining the patients’ labeled medical images, several machine learning algorithms were applied.

Supervised machine learning algorithms received a set of instances, which were individually labeled. Figure 1 shows the several steps belonging to the proposed method for the detection of pulmonary nodules.

As shown in Figure 1, the proposed method is composed of two distinct phases: *training* and *testing*. The *training* phase relates to the creation of a model (starting from a set of data called the *training* set), whereas the *testing* phase has the purpose of evaluating the effectiveness of the model learned in the *training* phase.

A dataset composed of different medical exam results belonging to 71 patients, 46 of whom were identified as suffering from malign pulmonary nodules and 25 of whom were detected as not suffering from malign pulmonary nodules, was gathered.

The proposed method started with the extraction of a set of radiomic features belonging to five different categories, as described in the previous subsection. In particular, for each medical image, the radiomic features were extracted with a script developed by authors invoking the PyRadiomics library.

Three different decision tree-based algorithms were applied: functional tree, Rep Tree and J48.

Decision tree-based algorithms use multiple algorithms to decide to split a node into two or more sub-nodes. The creation of sub-nodes increases the homogeneity of the resultant sub-nodes. Consequently, the purity of the node increases with respect to the target variable. The decision tree splits the nodes on the basis of all available variables and then selects the split which results in the most homogeneous sub-nodes.

In detail, the functional tree algorithm is aimed at building functional trees for classification, with particular regard to functional trees with logistic regression functions at the inner nodes and/or leaves. Basically, a functional tree is a diagram showing the dependencies between functions in a system (represented by the features in the case of machine learning), which is constructed with the aim of breaking down a problem into simpler parts. The Rep Tree algorithm builds a decision tree using information variance and prunes it using reduced-error pruning. It is considered to be an extension of the J48 supervised classification algorithm, improving the pruning phase by using reduced-error pruning (REP). The method uses a separate pruning dataset. For every subtree, it checks whether the subtree could be replaced by a single node without lowering the performance of the classifier on this pruning set. As such, the pruning method is simple, but is often considered to be too aggressive, i.e., it might remove subtrees which are actually relevant. The main difference between the Rep Tree algorithm and the J48 algorithm is the fact that J48 does not contain the REP step.

For model building, the Weka data science suite was exploited [16].

The proposed method was evaluated via k-cross validation, with k equal to 10.

### 2.4. Experimental Analysis

In the classification analysis, we considered four different metrics: precision, recall, the F-measure and accuracy.

Precision was computed as the proportion of the examples that truly belonged to class X among all those assigned to the class. It was expressed as the ratio of the number of relevant records retrieved to the total number of irrelevant and relevant records retrieved:$$\mathrm{Precision}=\frac{tp}{\left(tp+fp\right)}$$
where *tp* indicates the number of true positives and *fp* indicates the number of false positives.

The recall was computed as the proportion of examples that were assigned to class X among all the examples that truly belonged to the class, i.e., how much of the class was captured. Therefore, recall was expressed as the ratio of the number of relevant records retrieved to the total number of relevant records:$$\mathrm{Recall}=\frac{tp}{tp+fn}$$
where *tp* indicates the number of true positives and *fn* indicates the number of false negatives.

The F-measure is a measure of a test’s accuracy. This score can be interpreted as a weighted average of the precision and recall:$$F-Measure=2\frac{Precision\;\times \;Recall}{Precision+Recall}$$

The accuracy indicates how many times the model has correctly performed a classification compared to the total number of evaluated instances.

The receiver operating characteristic (ROC) area is a value that illustrates the diagnostic ability of a binary classifier system as its discrimination threshold is varied. The ROC curve was created by plotting the true positive rate against the false positive rate at various threshold settings.

## 3. Results

### 3.1. Clinical Results

The clinical and pathological features of the study group are reported in Table 2.

A total of 71 patients aged over 75 years, with a median age of 79 (interquartile range (IQR) 76–81), was analyzed. We found that 63.4% of them (45/71) were male and most of them were former smokers (44/71, 65.7%) or current smokers (10/71, 14.9%).

Patients had a median of three comorbidities (IQR 2–5) and the most common ones were hypertension (51/71, 71.8%) and chronic obstructive pulmonary disease (36/71, 50.7%). Thirty of them (42.2%) had a history of previous malignancy.

The median Charlson comorbidity index was six (IQR 5–7%), which was related to a 10-year survival of 2%. Regarding post-operative mortality and morbidity scores, the Thorascore and the POSSUM score, which are applied in lung surgery, were used. Median post-operative estimated mortality, according to the Thorascore, was 2.8% (IQR 1.2–6.3%). With reference to POSSUM, median post-operative mortality was 4.2% (3.1–8.5%), whereas post-operative estimated morbidity was 23.9% (IQR 17.7–44.3%).

The most frequent surgical procedure was wedge resection (41/71, 57.7%), followed by lobectomy (25/71, 35.2%), segmentectomy (4/71, 5.6%) and pneumonectomy (1/71, 1.5%). Wedge resection was often preferred due to the several comorbidities of the patients and their difficulty in tolerating single-lung ventilation for a long time.

At final histology, 35.2% of nodules (25/71) were benign, thus implying that in these cases surgery would have been avoidable.

Median hospitalization was eight (IQR 6–12) days. No post-operative mortality was recorded within 30 days from the surgical procedure. The post-operative morbidity rate was 31.0% (22/71). The main complications were prolonged anemia with the need of blood bank products (n = 7), pneumonia (n = 6), atrial fibrillation (n = 4), acute respiratory distress syndrome (n = 3; one patient required tracheostomy), prolonged air leaks (n = 3) and renal failure (n = 2).

### 3.2. Results of AI-Integrated Radiomic Analysis

The results of the experimental analysis are shown in Table 3. For the functional tree model we obtained an accuracy of 0.93 for healthy patients and an accuracy of 0.88 for patients with disease; thus, the average accuracy was 0.905. With regard to the Rep Tree model, we obtained an accuracy of 0.769 for healthy patients and 0.887 for patients with disease, and the average accuracy was 0.828. With the last model, i.e., J48, the accuracy was 0.761 for healthy patients and 0.889 for patients with disease, with an average accuracy of 0.825. All three different models analyzed proved satisfactory in predicting malignancy, although the best results in differentiating benign SPNs from malignant SPNs on the basis of radiomics data were obtained by the J48 model, as shown by the greater ROC area (Figure 2, Figure 3 and Figure 4).

## 4. Discussion

The presence of asymptomatic SPNs with the suspicion of malignancy is a challenging clinical scenario for the thoracic surgeon. Although such lesions are usually benign, the risk of early-stage and potentially curable malignant disease remains significant. Proper assessment of the nature of the SPN plays an important role in the therapeutic process of early lung cancer. Histologic confirmation of malignancy is usually required before anatomic lung resection (i.e., lobectomy), and this should be performed with either preoperative or intraoperative biopsy.

Considering CT section thickness is important in order to ensure the accuracy of nodule measurements. Several authors demonstrated that variability decreased with decreasing section thickness [17,18,19] and that the thinnest sections (usually 1 mm) provided the most consistent results [20].

Computed tomography (CT)-guided needle biopsy is challenging for smaller or more central nodules, is associated with high false-negative results and is burdened with a considerable rate of complications. Bronchoscopic sampling has a low yield for peripheral and small nodules, whereas intraoperative frozen section analysis may be difficult for small or central nodules and it can increase surgical time and cost. Alternatively, upfront surgery has been proposed for SPNs that are highly suspicious for NSCLC based on their clinical and imaging characteristics in the absence of tissue confirmation [21,22,23]. However, there is concern associated with performing unnecessary surgery for benign lesions and subjecting patients to potential morbidity.

Elderly patients (>75 years of age) represent a large segment of the affected population. Various studies have shown that the risk of post-operative mortality in this subset is equal to twice that of the population aged 65–69 years, or even threefold that of the general population [24,25]. The incidence of postoperative complications in certain studies reaches 48% in octogenarians [26]. Therefore, the definition of SPNs is of paramount importance in the diagnostic workup of such frail patients. In our study, 71 patients aged over 75 years (median 79, IQR 76–81) with SPNs, with a median of three comorbidities (IQR 2–5), were considered as the sample and the postoperative complication rate was 31.0%.

The use of AI analysis of radiomic features was applied to distinguish between benign and malignant nodules in order to create a model with the aim of reducing unnecessary surgery in elderly frail patients with suspicious SPNs. The predictive model for SPNs was based on 71 radiomic features belonging to five different classes: first-order features, shape, GLCM, GLRLM and GLSZM. The use of different classes and different combinations of radiomic features allowed for better diagnostic performance.

Amongst the three machine learning algorithms used, the performance of J48 was satisfactory, with a ROC area of 0.932, the largest compared to the functional tree and Rep Tree models, which had ROC areas of 0.847 and 0.915, respectively. J48 is one of the best machine learning algorithms to use when analyzing data with clear and easily understandable rules. The model generates categorical and continuous results and provides competitive performance.

The obtained results were consistent with those of several studies in the literature. Albano et al. showed the optimal accuracy of radiomic and PET-CT metabolic features, with an area under the curve (AUC) >0.8, compared to invasive procedures, in predicting the diagnosis of SPNs in 202 patients [27]. However, they used different criteria in selecting their sample compared to this study, considering patients >18 years of age without a previous history of any malignancy, surgery, chemotherapy or radiotherapy and considering 42 radiomic features instead of 71.

Other studies have shown an AUCROC >0.70 in the use of such methodologies to discern nodules using CT or PET-CT data. Kumar et al. obtained an accuracy of 79.06%, whereas Liu et al. achieved an accuracy of 81% and Wu et al. obtained an accuracy of 72% [28,29,30]. In addition, Niu et al. showed how adding SUV_max_ to the CT radiomics analysis improves its predictive value in differentiating between benign and malignant ground-class nodules, with a diagnostic efficiency of 0.940 [31].

Astaraki et al. conducted a comparison between the performance of radiomics and deep-learning models for SPN malignancy prediction [32]. The database included 1297 nodules. With the classical radiomics approach, 102 features were extracted and their predictive power was subsequently analyzed through the use of eight machine learning algorithms. In this case, the use of decision trees yielded an AUCROC of 0.723 ± 0.011, whereas adaptive boosting embedded on decision trees yielded the highest prediction power, with an AUCROC of 0.889 ± 0.016, among all the used algorithms.

Parmar et al. focused on different classification models, reporting a variation in performance of 34.21% depending on the system used [33]. They evaluated 440 radiomic characteristics in 464 images and used 12 different classifiers belonging to different classes. In this case, however, the use of decision trees produced unsatisfactory results, with an AUCROC of 0.54 ± 0.04 compared to the random forest method because of the latter’s high stability against data perturbation.

These results are consistent with the idea that it is possible to obtain a safe indication in a completely non-invasive way, without proceeding to a pre-surgical biopsy or an avoidable resection. This in line with studies such as that of Ghamati et al., who showed that it is common practice to carry out a lung resection without a preoperative defined diagnosis, risking an increase in the rates of unnecessary surgery [34]. The physician’s ability to make a reliable lung cancer diagnosis solely based on clinical and radiologic data in fact represents an undeniable challenge.

Radiomics has proven to be a fundamental tool in reaching this goal. Indeed, according to a review by Senent-Valero et al., many predictive models based on clinical and radiological characteristics, without the aid of radiomics, have displayed numerous pitfalls [35]. Most of these are based on retrospective studies and a low level of methodological rigor. On the other hand, Zhang et al. considered both clinical and radiomic predictors in the evaluation of pulmonary malignancy nodules, showing a higher predictive value than those mentioned by Senent-Valero et al. (AUC 0.89–0.91 vs. 0.59–0.70) [36]. Furthermore, Zhao et al. obtained a higher model accuracy (81%) for the method based on radiomic characteristics than the conventional CT model (63.7%) [37]. Rafael-Palou et al. [38] proposed a method aiming to detect lung nodules by exploiting neural networks. The main difference of this method with respect to the proposed method is the introduction of radiomic features in order to discriminate between healthy and disease-affected patients, whereas neural networks automatically extract features from images under analysis.

Zheng and colleagues [39] adopted an approach similar to the one proposed by the authors in [38], yet they considered the task of automatically segmenting pulmonary nodules. Conversely, in this paper, we have proposed a method aimed to detect the presence of cancer using the whole image (i.e., the proposed method does not require the user to manually segment a set of images for model training). Although the proposed method achieved a very good performance, the present study represents a retrospective, single-institution, nonrandomized experience with a small sample size and a referral center bias. In future works, it might be necessary to increase the training and validation datasets in order to deal with variability in SPN morphology, as well as to design a prospective study.

## 5. Conclusions

Due to the rapid improvements in artificial intelligence (AI), and particularly in machine learning (ML), these methods have a wide range of clinical applications in lung cancer imaging, and are used by a growing number of radiologists, as witnessed by recently published series and surveys [40,41,42,43,44]. Limitations in the development of new ML tools is mainly due to the difficulties in recruitment and the availability of imaging data.

Radiomics has proven to be a promising method of diagnosis in early lung cancer. It can be useful not only in cancer detection and staging, but also in predicting the response to therapy, with a wide field of applicability.

Furthermore, in support of the results of previous studies, decision trees seem to be the best method of analysis of radiomic features, achieving the greatest prognostic performance. These tools can be a valid alternative to invasive diagnostic procedures in the decision-making process and in the management of elderly patients with SPNs that are suspicious for early-stage lung cancer, and finally reduce the rate of “unnecessary” surgical procedures. In our study, the J48 model showed the best performance compared to the functional tree and Rep Tree models.

However, the long-term impact of such techniques on patient selection in terms of the best treatment strategy, final outcomes and cost/benefit ratios is still not clear. The use of open-source tools for algorithm development, where possible, is warranted to improve the diagnostic performance of all AI software algorithms.

## Figures and Tables

**Figure 1 diagnostics-13-00384-f001:**
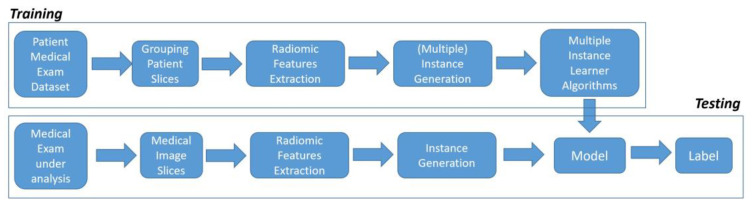
The proposed method for pulmonary nodule detection.

**Figure 2 diagnostics-13-00384-f002:**
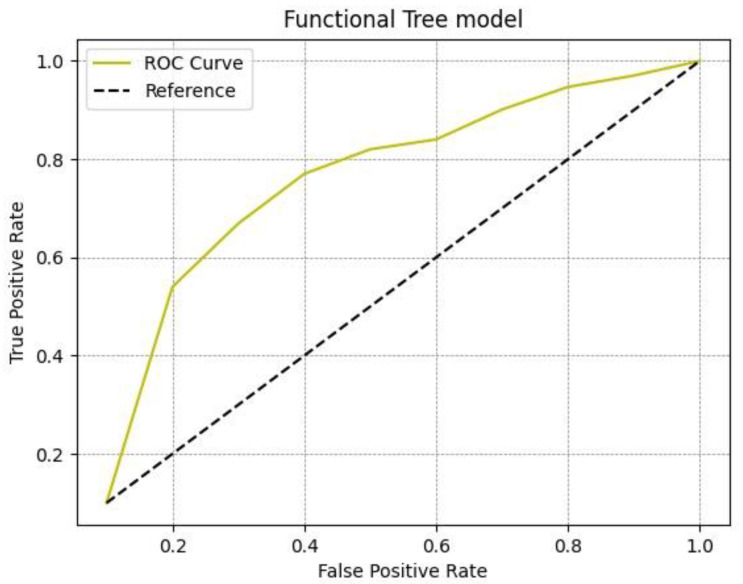
ROC curve for the functional tree model.

**Figure 3 diagnostics-13-00384-f003:**
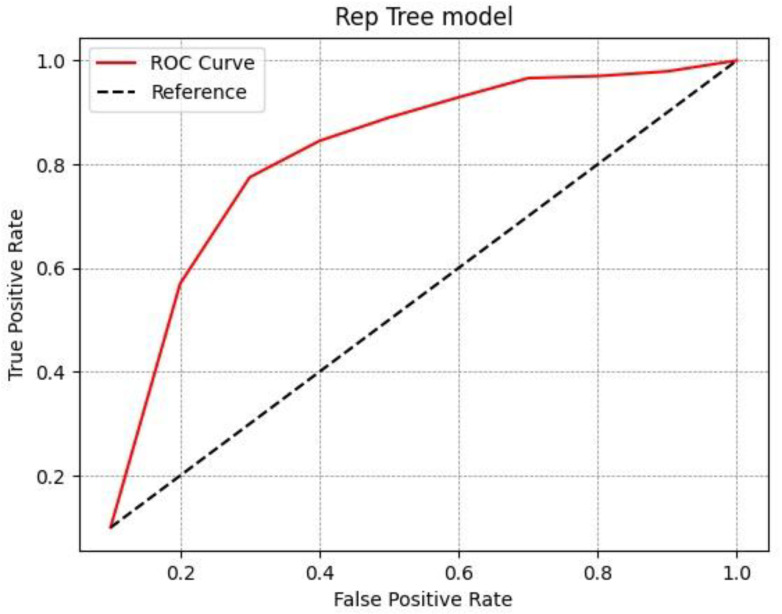
ROC curve for the Rep Tree model.

**Figure 4 diagnostics-13-00384-f004:**
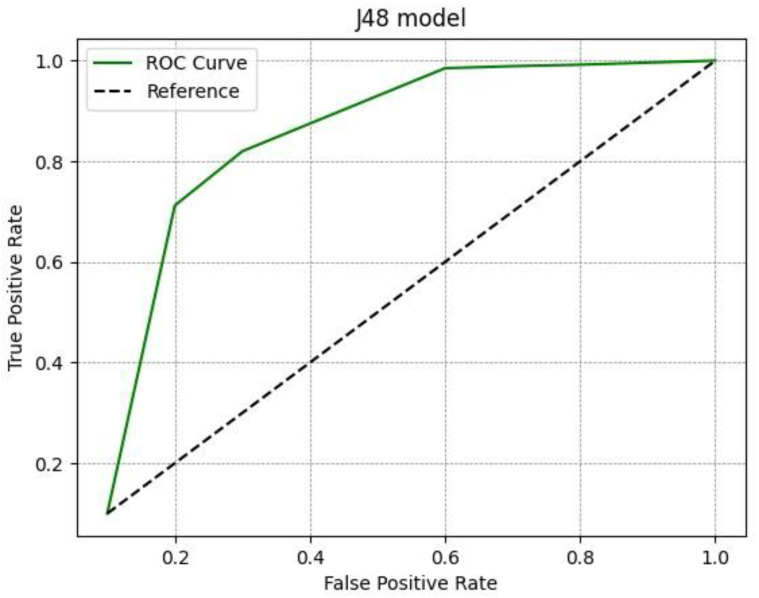
ROC curve for the J48 model.

**Table 1 diagnostics-13-00384-t001:** Radiomic features considered in this study.

1	Class First Order	Feature Mean	Description ROI Average Gray Intensity
2	Shape	Elongation	relationship between two largest principal components
3	Shape	Flatness	relationship between largest and smallest principal components
4	Shape	LeastAxisLength	yield smallest axis length of the ROI-enclosing ellipsoid
5	Shape	MajorAxisLength	yield largest axis length of ROI-enclosing ellipsoid
6	Shape	Maximum2DDiameterColumn	mesh vertices in row-slice plane
7	Shape	Maximum2DDiameterRow	mesh vertices in the column-slice plane
8	Shape	Maximum2DDiameterSlice	mesh vertices in row-column plane
9	Shape	Maximum3DDiameter	mesh vertices
10	Shape	MeshVolume	volume is obtained using the surface mesh
11	Shape	MinorAxisLength	second-largest axis length of the ROI-enclosing ellipsoid
12	Shape	Sphericity	roundness of shape of the tumor region relative to a sphere
13	Shape	SurfaceArea	the sum of all sub-areas
14	Shape	SurfaceVolumeRatio	Surface Area to Volume ratio
15	Shape	VoxelVolume	approximate volume
16	GLCM	Autocorrelation	magnitude of the fineness and coarseness of texture
17	GLCM	ClusterProminence	skewness and asymmetry of the GLCM
18	GLCM	ClusterShade	skewness and uniformity of the GLCM
19	GLCM	ClusterTendency	voxels with similar gray-level values
20	GLCM	Contrast	the local intensity variation
21	GLCM	Correlation	linear dependency of gray-level values
22	GLCM	DifferenceAverage	occurrences of pairs with similar and differing intensity values
23	GLCM	DifferenceEntropy	randomness/variability in neighborhood intensity value differences
24	GLCM	DifferenceVariance	heterogeneity of higher weights on differing intensity level pairs
25	GLCM	Id	inverse difference
26	GLCM	Idm	inverse difference moment
27	GLCM	Idmn	Inverse difference Moment Normalized
28	GLCM	Idn	Inverse difference Normalized
29	GLCM	Imc1	informational measure of correlation 1
30	GLCM	Imc2	informational measure of correlation 2
31	GLCM	InverseVariance	inverse of the variance
32	GLCM	JointAverage	the mean gray-level intensity of the distribution
33	GLCM	JointEnergy	a measure of homogeneous patterns in the image
34	GLCM	JointEntropy	measure of the randomness/variability in neighborhood intensity values
35	GLCM	MCC	maximal correlation coefficient
36	GLCM	MaximumProbability	occurrences of the most predominant pair of neighboring intensity values
37	GLCM	SumAverage	occurrences of pairs with lower and higher intensity values
38	GLCM	SumEntropy	sum of neighborhood intensity value differences
39	GLCM	SumSquares	distribution of neigboring intensity level pairs
41	GLRLM	GLN	gray-level non-uniformity
42	GLRLM	GLNN	gray-level non-uniformity normalized
43	GLRLM	GLV	gray-level variance
44	GLRLM	HGLRE	high gray-level run emphasis
45	GLRLM	LRE	long run emphasis
46	GLRLM	LRHGLE	long run high gray-level emphasis
47	GLRLM	LRLGLE	long run low gray-level emphasis
48	GLRLM	LGLRE	low gray-level run emphasis
49	GLRLM	RE	run entropy
50	GLRLM	RLN	run length non-uniformity
51	GLRLM	RLNN	run length non-uniformity normalized
52	GLRLM	RP	run percentage
53	GLRLM	RV	run variance
54	GLRLM	SRE	short run emphasis
55	GLRLM	SRHGLE	short run high gray-level emphasis
56	GLRLM	SRLGLE	short run low gray-level emphasis
57	GLSZM	GLN	gray-level non-uniformity
58	GLSZM	GLNN	gray-level non-uniformity (normalized)
59	GLSZM	GLV	gray-level variance
59	GLSZM	HGLZE	high gray-level zone emphasis
60	GLSZM	LAE	large area emphasis
61	GLSZM	LAHGLE	large area high gray-level emphasis
62	GLSZM	LALGLE	large area low gray-level emphasis
63	GLSZM	LGLZE	low gray-level zone emphasis
64	GLSZM	SZN	size-zone non-uniformity
65	GLSZM	SZNN	size-zone non-uniformity normalized
66	GLSZM	SAE	small area emphasis
67	GLSZM	SAHGLE	small area high gray-level emphasis
68	GLSZM	SALGLE	small area low gray-level Emphasis
69	GLSZM	ZE	zone entropy
70	GLSZM	ZP	zone percentage
71	GLSZM	ZV	zone variance

ROI = region of interest; GCLM = gray-level co-occurrence matrix; GLRLM = gray-level run length Matrix; GLSZM = gray-level size zone matrix.

**Table 2 diagnostics-13-00384-t002:** Characteristics of the enrolled patients.

Variable	
Median age, years (IQR)	79 (76–81)
Gender, n (%)	
Male Female	45 (63.4%) 26 (36.6%)
Median number of comorbidities (IQR)	3 (IQR 2–5)
Cardiac comorbidities, n (%) Coronary artery disease Any previous cardiac surgery Current treatment for hypertension Current treatment for arrhythmia Current treatment for cardiac failure	18 (25.3%) 13 (18.3%) 51 (71.8%) 15 (21.1%) 8 (11.3%)
Non-cardiac comorbidities, n (%) Insulin-dependent diabetes Serum creatinine >2 mg/dL Cerebral vascular accident Chronic kidney failure Chronic obstructive pulmonary disease Gastric ulcer Liver disease Previous malignancy Other comorbidities	19 (26.8%) 2 (2.8%) 8 (11.3%) 6 (8.4%) 36 (50.7%) 8 (11.3%) 8 (11.3%) 30 (42.2%) 35 (49.3%)
Smoking history, n (%) Never smoked Former smoker Current smoker Unknown	12 (17.9%) 44 (65.7%) 10 (14.9%) 1 (1.5%)
Median Charlson Comorbidity Index (IQR)	6 (5–7)
Median Thorascore (IQR)	2.8% (1.2–6.3%)
Median POSSUM mortality (IQR)	4.2% (3.1–8.5%)
Median POSSUM morbidity (IQR)	23.9% (17.7–44.3%)
Surgical procedure, n (%) Pneumonectomy Lobectomy Segmentectomy Wedge resection	1 (1.5%) 25 (35.2%) 4 (5.6%) 41 (57.7%)
Final histology, n (%) Lung adenocarcinoma Lung squamous carcinoma Lung metastasis Typical carcinoid Atypical carcinoid Benign lesion	30 (42.3%) 11 (15.5%) 1 (1.4%) 2 (2.8%) 2 (2.8%) 25 (35.2%)
30-day post-operative mortality, n (%)	0 (0.0%)
Post-operative morbidity, n (%)	22 (31.0%)
Median hospitalization, days (IQR)	8 (6–12)

IQR: interquartile range.

**Table 3 diagnostics-13-00384-t003:** Experimental analysis results.

Model	Precision	Recall	F-Measure	ROC Area	Accuracy	Label
Functional Tree	0.88	1	1	0.94	0.93	Healthy
	0.866	0.913	0.889	0.847	0.88	Disease
	0.854	0.855	0.854	0.847	0.905	weighted avg.
Rep Tree	0.836	0.741	0.786	0.915	0.769	Healthy
	0.861	0.916	0.888	0.915	0.887	Disease
	0.852	0.853	0.850	0.915	0.828	weighted avg.
J48	0.854	0.740	0.792	0.932	0.761	Healthy
	0.861	0.927	0.893	0.932	0.889	Disease
	0.858	0.859	0.856	0.932	0.825	weighted avg.

## Data Availability

Not applicable.

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
