# Peer review of "Radiomics and Artificial Intelligence Can Predict Malignancy of Solitary Pulmonary Nodules in the Elderly"

_diagnostics, 2023, doi:10.3390/diagnostics13030384_

Round 1
Reviewer 1 Report
This article is interesting in that the pulmonary nodules with histological diagnosis are retrospectively assessed using the radiomics. However, I think that there is room for improvement, and recommend the authors for major revision.
Major:
1) The reason why the authors selected patients with solitary pulmonary nodules (SPNs) aged 75 years or older remains unclear. This method seems applicable to younger patients.
2) The authors should clearly state the inclusion criteria of the patients with SPN in this study, in other words, whether they were chosen arbitrary or not. I think the inclusion criteria is very important in this retrospective study, and the authors did not mention that “consecutive patients with SPN” in the “Patients Selection” subsection.
3) The authors explained in the “Materials and Methods” section that CT “or” PET/CT scans of patients who had undergone resection of SPN were assessed in this study. I think the spatial resolution of a CT image surpasses that of FDG-PET/CT, because PET/CT data are usually acquired under the shallow breathing. Thus, the authors should explain the effect of evaluating SPNs only by PET/CT scans in some patients in this study.
4) The CT slice thickness is also an important factor in the evaluation of SPNs. Therefore, the authors should explain this point in the “Materials and Methods” section, and also the “Discussion” section.
Minor:
1) The difference between the three “decision tree-based algorithms” (Functional Tree, Rep Tree, J48) should be explained in the “Materials and Methods” section, in order to facilitate the potential readers’ understanding. Although the authors added a simple expression of J48 in the “Discussion” section (Lines 252-254), I think this is not good enough.
2) The number of patients in Table 1 is different from the 71 patients described in the manuscript.
3) The abbreviations used in Table 1 should be defined in the “Materials and Methods” section.
4) There are lots of grammatical errors in the manuscript, and the manuscript should be checked by a native speaker.
Author Response
We thank the reviewers for the time spent in reviewing our manuscript and hope we addressed all required changes and suggestions.
We enclose a two column word file with the list of required actions on the left and the taken action on the right for better comparison.

Reviewer 2 Report
-Could the title be made more concise?
-Explicit contributions of the paper needs to be fleshed out
-Methods;... is it just the 2 patients?
-Figure 1 requires improvements
-Could the features be put through other Machine Learning models?
-The results may require the "Accuracy" metric if not present already
-Conclusion requires substantial bolstering in addition to a pathway for future work
-Refs need some expansion
Author Response

(The authors gave the same response as above.)

Round 2
Reviewer 1 Report
The manuscript has been well revised, reflecting the reviewer’s comments.
Reviewer 2 Report
Thanks for making the changes.